# The Ycf48 accessory factor occupies the site of the oxygen-evolving manganese cluster during photosystem II biogenesis

Ziyu Zhao [1,6], Irene Vercellino [2,5,6], Jana Knoppová [3,6], Roman Sobotka [3,4], James W. Murray [1], Peter J. Nixon [1,7] ✉, Leonid A. Sazanov [2,7] ✉ & Josef Komenda [3,4,7] ✉

Robust oxygenic photosynthesis requires a suite of accessory factors to ensure efficient assembly and repair of the oxygen-evolving photosystem two (PSII) complex. The highly conserved Ycf48 assembly factor binds to the newly synthesized D1 reaction center polypeptide and promotes the initial steps of PSII assembly, but its binding site is unclear. Here we use cryo-electron microscopy to determine the structure of a cyanobacterial PSII D1/D2 reaction center assembly complex with Ycf48 attached. Ycf48, a 7-bladed beta pro-peller, binds to the amino-acid residues of D1 that ultimately ligate the water-oxidising $Mn_4CaO_5$ cluster, thereby preventing the premature binding of $Mn^{2+}$ and $Ca^{2+}$ ions and protecting the site from damage. Interactions with D2 help explain how Ycf48 promotes assembly of the D1/D2 complex. Overall, our work provides valuable insights into the early stages of PSII assembly and the structural changes that create the binding site for the $Mn_4CaO_5$ cluster.

Photosystem II (PSII) is the multi-subunit membrane-bound pigment-protein complex found in oxygenic photosynthetic organisms that catalyses the light-driven oxidation of water and reduction of plasto-quinone during photosynthetic electron transport[1]. PSII is a major biological route for the conversion of solar energy into chemical energy and its activity is vital for the growth of plants, algae and cya-nobacteria and the production of oxygen that is necessary for aerobic life[2]. PSII is also a weak link in photosynthesis due to its vulnerability to irreversible photoinhibition, and damaged subunits are replaced through an elaborate repair cycle involving the disassembly and reassembly of PSII[3].

Assembly of PSII in cyanobacteria occurs sequentially from smaller pigment-protein sub-complexes or modules[3,4] and involves the participation of several accessory factors to optimise and regulate assembly and to protect the nascent complexes from photoinhibitory

damage[3,5]. Recent advances in cryo-electron microscopy (cryo-EM) have yielded the structures of a variety of non-oxygen-evolving pre-cursor PSII complexes, allowing detailed structural insights into the assembly of PSII and the roles of associated accessory factors such as Psb27[6,7], Psb28 and Psb34[7,8].

The Ycf48 accessory factor (known as HCF136 in chloroplasts) plays an important role in cyanobacteria[9] and chloroplasts[10,11] at an early stage of PSII assembly. Ycf48 binds to unassembled precursor and mature forms of the D1 reaction center (RC) subunit involved in binding the chlorophyll (Chl), plastoquinone and metal ion cofactors essential for PSII function. Once attached to D1, Ycf48 promotes the formation of PSII reaction centre assembly complexes (RCII) from the D1 module (D1mod) containing D1 and the PsbI subunit and the D2 module (D2mod) containing the D2 reaction center subunit and cyto-chrome (Cyt) b-559[9,12] (Fig. 1). The assembly of PSII then proceeds by

[1]Sir Ernst Chain Building-Wolfson Laboratories, Department of Life Sciences, Imperial College London, S. Kensington Campus, London SW7 2AZ, UK. [2]Institute of Science and Technology Austria, 3400 Klosterneuburg, Austria. [3]Institute of Microbiology, Academy of Sciences of the Czech Republic, Opatovický mlýn, Třeboň 379 81, Czech Republic. [4]Faculty of Science, University of South Bohemia, Branišovská 31, České Budějovice 37005, Czech Republic. [5]Present address: Forschungszentrum Jülich GmbH, Wilhelm-Johnen-Straße, 52428 Jülich, Germany. [6]These authors contributed equally: Ziyu Zhao, Irene Vercellino, Jana Knoppová. [7]These authors jointly supervised this work: Peter J. Nixon, Leonid A. Sazanov, Josef Komenda. ✉e-mail: p.nixon@imperial.ac.uk; sazanov@ist.ac.at; komenda@alga.cz

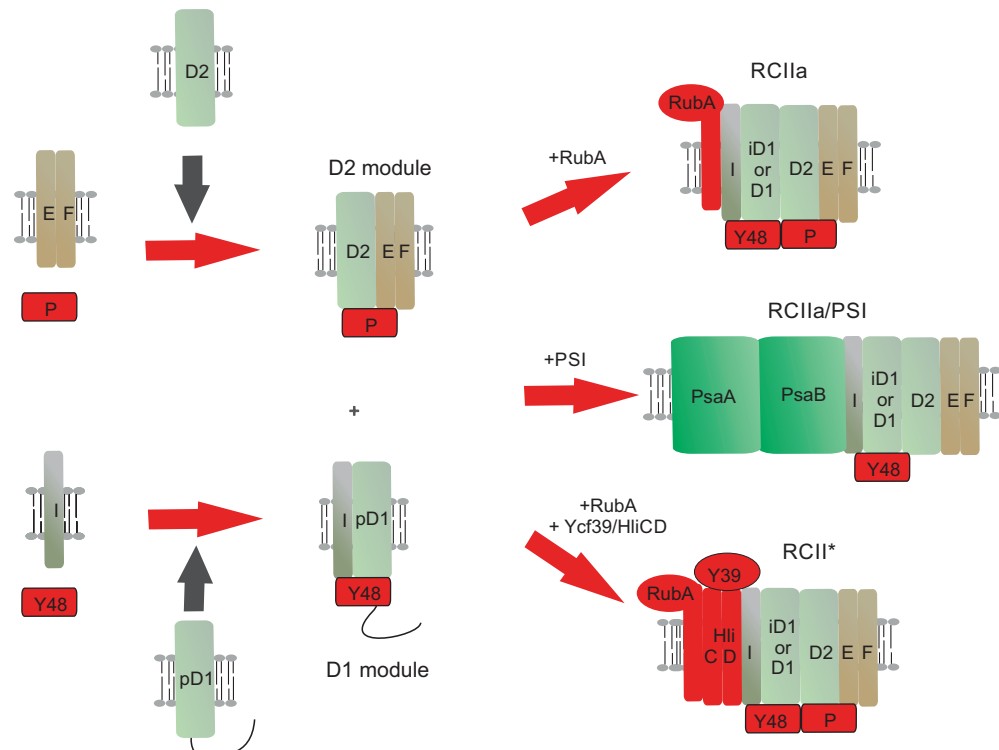

**Fig. 1 | Simple model of the initial stages of assembly of PSII and the various isolated RCII complexes.** The D1 module contains precursor D1 (pD1), PsbI (I) and Ycf48 (Y48) and the D2 module contains D2, the PsbE (E) and PsbF (F) subunits of Cyt b-559 and CyanoP (P). Precursor D1 is cleaved to intermediate D1 (iD1) and mature D1 upon formation of the RCII complexes, RCIIa and RCII*. RCII* is equivalent to RCIIa but additionally contains the Ycf39 (Y39)/Hlips (High light inducible proteins) complex. The Hlips complex (HliCD) consists of the HliC and HliD subunits. The rubredoxin-like RubA protein is found in RCIIa and RCII*. The His-tagged RCII/PSI complex studied here was purified by immobilised $Ni^{2+}$-affinity chromatography followed by sucrose density gradient centrifugation. Only the large PsaA and PsaB RC subunits are displayed in PSI for clarity.

attachment of the CP47 module to form the RC47 complex followed by the CP43 module to form a monomeric PSII core complex (Supplementary Fig. 1)[3,4,13]. The final maturation steps involve light-driven assembly of the oxygen-evolving $Mn_4CaO_5$ cluster in a binding site created by amino-acid residues of the D1 and CP43 subunits[14], attachment of extrinsic proteins to the lumenal surface of PSII[15] and dimerization of the complex[3].

Ycf48 is present at stoichiometric levels in isolated RCII complexes[12] and at lower levels in the RC47 complex[16] but is absent from oxygen-evolving complexes. Besides a role in PSII assembly, Ycf48 is also important for the repair of photodamaged PSII complexes, most likely by stabilising newly synthesised D1 needed for the selective replacement of damaged D1[9].

Ycf48 is a 7-bladed beta propeller protein possessing a highly conserved cluster of Arg residues (so-called 'Arg patch'), which muta-genesis has shown to be important for binding to RCII[16]. We have previously suggested that the Arg patch might bind to one or more of the carboxylate residues on the lumenal surface of D1 that ligate the $Mn_4CaO_5$ cluster[16]. Here we have used cryo-EM to determine the binding site of Ycf48 in a RCII assembly complex which was isolated attached to a monomeric PSI complex[12]. Our results reveal that Ycf48 binds onto the surface of D1 through the Arg patch, at the site where the oxygen-evolving Mn cluster binds to D1 in mature PSII. Additionally, the C-terminal tail of D1, involved in ligating the Mn cluster in mature PSII, is wrapped around Ycf48. Consequently, detachment of Ycf48 is an obligatory step for light-driven assembly of the $Mn_4CaO_5$ cluster. Our work provides insights into the role of Ycf48 in the bio-genesis and repair of PSII and the regulation of assembly of the oxygen-evolving complex of photosynthesis.

## Results

### Purification and properties of a RCII/PSI complex

We have previously described the purification of photoactive RCII assembly complexes from the cyanobacterium *Synechocystis* sp. PCC 6803 (hereafter Syn6803)[12]. Our approach involved isolating His-tagged D2 complexes from a mutant lacking CP47 that is blocked in PSII assembly so that only RCII assembly complexes accumulate[12]. In addition to the isolation of two types of RCII complex (RCII* and RCIIa), a larger 450 kDa complex consisting of the RCIIa complex bound to a monomeric PSI complex, which we term here the RCII/PSI complex, was also detected in the preparation (Fig. 1). We used an additional sucrose density gradient centrifugation step to purify this complex (Supplementary Fig. 2). SDS-PAGE analysis coupled to immunoblotting and mass spectrometry confirmed that the isolated complex con-tained the PsaA, PsaB, PsaC, PsaD, PsaE, PsaF, PsaL, PsaK1 and PsaK2 subunits of PSI, the PSII subunits D1, D2, PsbI, PsbE and PsbF of Cyt b-559 plus the Ycf48 accessory factor (Supplementary Fig. 3, Supple-mentary Table 1). Both precursor and mature forms of D1 were detected in the RCII/PSI preparation suggesting a heterogeneous population of complexes. The rubredoxin-like RubA protein[17], Ycf39 and the Hlip subunit, HliD, reported previously to associate with RCII complexes[18] were also detected, but in lower amounts (Supplementary Table 1).

### Overall structure of the RCII/PSI complex

We collected 2853 cryo-EM micrographs of the RCII/PSI complex and picked 1.25 million particles using LoG picking[19] for subsequent data processing. The preliminary processing revealed the existence of two good 3D classes out of a total of six, differing by the relative orientation

of RCII to PSI (Supplementary Figs. 4 and 5). The particles corresponding to the two classes (106532 particles) were used as references to repeat picking using Topaz[20] and this new set of particles was subject to multiple rounds of 3D classification, as well as polishing and refinements. The best particles were finally separated into 3 classes by 3D classification without alignment, resulting in the same two good classes identified initially, but with more particles per class and higher resolution. The final cryo-EM maps had a resolution of 3.2 Å and 3.1 Å for the two final classes. Additionally, all the particles were pooled to generate a consensus PSI map at 2.9 Å. The data processing pipeline is explained in detail in the Methods section and outlined in Supplementary Fig. 4 and Supplementary Table 2 and examples of the modelling shown in Supplementary Figs. 6 and 7.

The overall structure obtained using the 3.1 Å map contains a monomeric PSI complex consisting of 11 subunits attached to a RCII complex consisting of the intrinsic D1 and D2 RC subunits, the low-molecular-mass PsbI subunit and the alpha (PsbE) and beta (PsbF) subunits of Cyt b-559, plus the Ycf48 accessory factor (Fig. 2). RubA, Ycf39 and the Hlip proteins were not resolved, presumably because of the low abundance of complexes containing these subunits. Unusually the PSI and RCII complexes were present in opposing orientations to that expected from their natural orientation in the thylakoid membrane so that the cytoplasmically exposed PsaC, PsaD and PsaE subunits of PSI[21] are on the same side of the complex as the lumenal Ycf48 factor (Fig. 2a). The RCII/PSI complex therefore appears to be artefactual, although we cannot totally exclude the possibility that the complex might reflect interactions at an early stage of PSII assembly within specialised but still ill-defined biogenesis regions of the membrane[22]. Recent work has also suggested that trimeric PSI complexes might exist in opposite orientations in the thylakoid membrane[23].

Examination of the interface revealed that RCII mainly interacts with PSI via hydrophobic interactions between residues D1-Phe295 and D1-Leu297 and residues PsaB-Phe161, PsaB-Leu155 and PsaB-Phe151 in the PSI PsaB RC subunit plus chlorophyll 1208, and by interactions between the C-terminus of PsbI and PsaB. Ycf48-Lys261 is in the vicinity of the backbone carbonyl oxygen of PsaB-Asn294 and PsbF-Phe41 lies close to PsaI-Leu29 (Supplementary Fig. 8).

## Cofactors within the RCII assembly complex

The cryo-EM structure of RCII revealed the presence of 6 Chl $a$ and 2 pheophytin $a$ (Pheo a) molecules within the D1/D2 heterodimeric complex plus the single heme of Cyt b-559 (Fig. 2b), in line with previous quantification of pigment content[12] and the known binding of pigment to this region of PSII[24–26]. The two beta rings of the carotenoid bound to D1 could be clearly identified in the structure (Fig. 2b, Supplementary Fig. 6b) but not those of the D2 carotenoid possibly because it was lost during purification or is present at lower occupancy in the assembly complex. Also present in the structure is the non-heme iron on the acceptor side of the complex. However, plastoquinones $Q_A$ and $Q_B$, which act as electron acceptors in oxygen-evolving PSII, could not be clearly identified (Fig. 2b). It is possible that they are not stably bound at this stage of assembly or are lost during purification so that photochemical activity is restricted to light-induced primary electron transfer between P680 and $Pheo_{D1}$[12], as observed for D1/D2 complexes isolated biochemically from detergent fragmentation of larger PSII core complexes[27].

The two redox-active tyrosine residues at D1-Tyr161 ($Tyr_Z$) and D2-Tyr160 ($Tyr_D$) and the associated histidine residues (D1-His190 and D2-His189) are in similar locations to that observed in larger oxygen-evolving complexes (Fig. 2b). The strength of the H-bond between $Tyr_Z$ and D1-His190 is crucial for rapid oxidation of $Tyr_Z$, the electron carrier that links oxidized P680+ (equivalent to $P_{D1}^+$ in Fig. 2b) to the Mn cluster[28]. The length of the H-bond was estimated to be 3.0 Å similar to the 3.2 Å distance in core complexes lacking the $Mn_4CaO_5$ cluster (PDB

ID: 6WJ6)[29] but longer than the 2.7 Å observed in Syn6803 oxygen-evolving complexes (PDB ID: 7N8O)[25] and the 2.5 Å reported for the 1.9 Å structure from *Thermosynechococcus vulcanus* (PDB ID:3WU2)[26]. D1-Asn298, which H-bonds to D1-His190 to promote $Tyr_Z$ oxidation, was in a similar position to that found in oxygen-evolving complexes but binding of Ycf48 likely disrupts the microenvironment around the D1-Tyr161/D1-His190/D1-Asn298 triad preventing efficient oxidation of $Tyr_Z$[30]. Importantly, there was no indication in the structure for the presence of bound Mn or Ca ions. Additionally, the two chloride ions, called Cl-1 and Cl-2, required for functionality of the oxygen-evolving complex[31] were not identified in the cryo-EM map although it should be noted that anions are challenging to resolve at medium resolution in cryo-EM[32]. The Cl-1 site close to D2-Lys317 in oxygen-evolving PSII clashes with the side chain of Ycf48-Arg196 and Cl-2 is occupied by the backbone of Ycf48-Thr223.

The four authentic His ligands to the non-heme iron (D1-His215 and His272 and D2-His214 and His268) could be identified in RCII (Supplementary Fig. 6b). However, the structures of both D1 and D2 around the non-heme iron (from D1-Arg225 to D1-Asn266 and D2-Phe223 to D2-Thr243) were poorly resolved suggesting structural flexibility (Supplementary Fig. 6a, Supplementary Table 1). It therefore remains unclear if the non-heme iron is also coordinated by bicarbonate, as in oxygen-evolving PSII[24,26], or by an alternative ligand, such as D2-Glu241, as observed recently in PSII assembly complexes[7,8] and Chl f-containing core complexes[33]. Other regions that could not be modelled in the RCII/PSI complex include the N- and C-terminal regions of PSII subunits and the N-terminal region of PsaK1 and the C-terminal tail of PsaL (Supplementary Fig. 6a, Supplementary Fig. 7a, Supplementary Table 1). Overall, the structures of the transmembrane regions of the PSII subunits in RCII were similar to that found in larger core complexes with only slight changes in the position of Cyt b-559 detected, which might alter the standard reduction potential of the heme[34].

## Interaction of Ycf48 with D1 and D2

Ycf48 in Syn6803 undergoes N-terminal processing followed by lipidation of the Ycf48-Cys29 N-terminal residue[35]. The first N-terminal residue of Ycf48 modelled in the structure was Ycf48-His31, located as expected close to the predicted surface of the membrane (Fig. 3b)[35].

We found that Ycf48 bound to the lumenal regions of the D1 subunit via an Arg patch consisting of six highly conserved residues (Ycf48-Arg180, Arg196, Arg215, Arg219, Arg220 and Arg237) (Fig. 3a–c)[16].

Residues Ycf48-Arg220 and Ycf48-Arg237 form salt bridges with residues D1-Asp170 and D1-Glu189, which act as ligands to the Mn cluster in the oxygen-evolving complex (Fig. 3c–e; Fig. 4) and might form the high-affinity binding site for the first Mn ion during photo-assembly[7,36–38] (Fig. 3e). Although relatively poorly resolved in the structure, Ycf48-Arg180 is located close to D1-Asp61 (Fig. 4b, d), which lies in the second coordination sphere of the Mn cluster and is involved in a proton channel in oxygen-evolving PSII[39]. The backbone oxygen of D1-Gly166 forms a 2.9 Å hydrogen bond with the backbone nitrogen of Ycf48-Gly238 (Fig. 4b, d) and potential weak hydrogen bonds (3.8 Å–4.5 Å) are present between Ycf48-Arg215 and D1-Asn296 (Fig. 4a), between Ycf48-Arg196 and D1-Asn181 (Fig. 4d) and between Ycf48-Arg219 and the backbone (carbonyl groups) of D1-Ile163, D1-Gly164 and D1-Gly166 (Fig. 4a, b).

The structure also revealed a dramatic change, starting from D1-Val330, in the structure of the C-terminal region of D1 which in oxygen-evolving PSII provides ligands at D1-His332, Glu333, Asp342 and the C-terminus at Ala344 to the $Mn_4CaO_5$ cluster (Fig. 3e). Instead of folding back close to D1-Asp170 and D1-Glu189, as observed in the RC47 complex and the mature PSII complex (Fig. 3c–e), the C-terminal tail is unfolded and wrapped around the outside of the Ycf48 propeller, with H-bonds formed between Ycf48 and D1-Asn335 and the last four

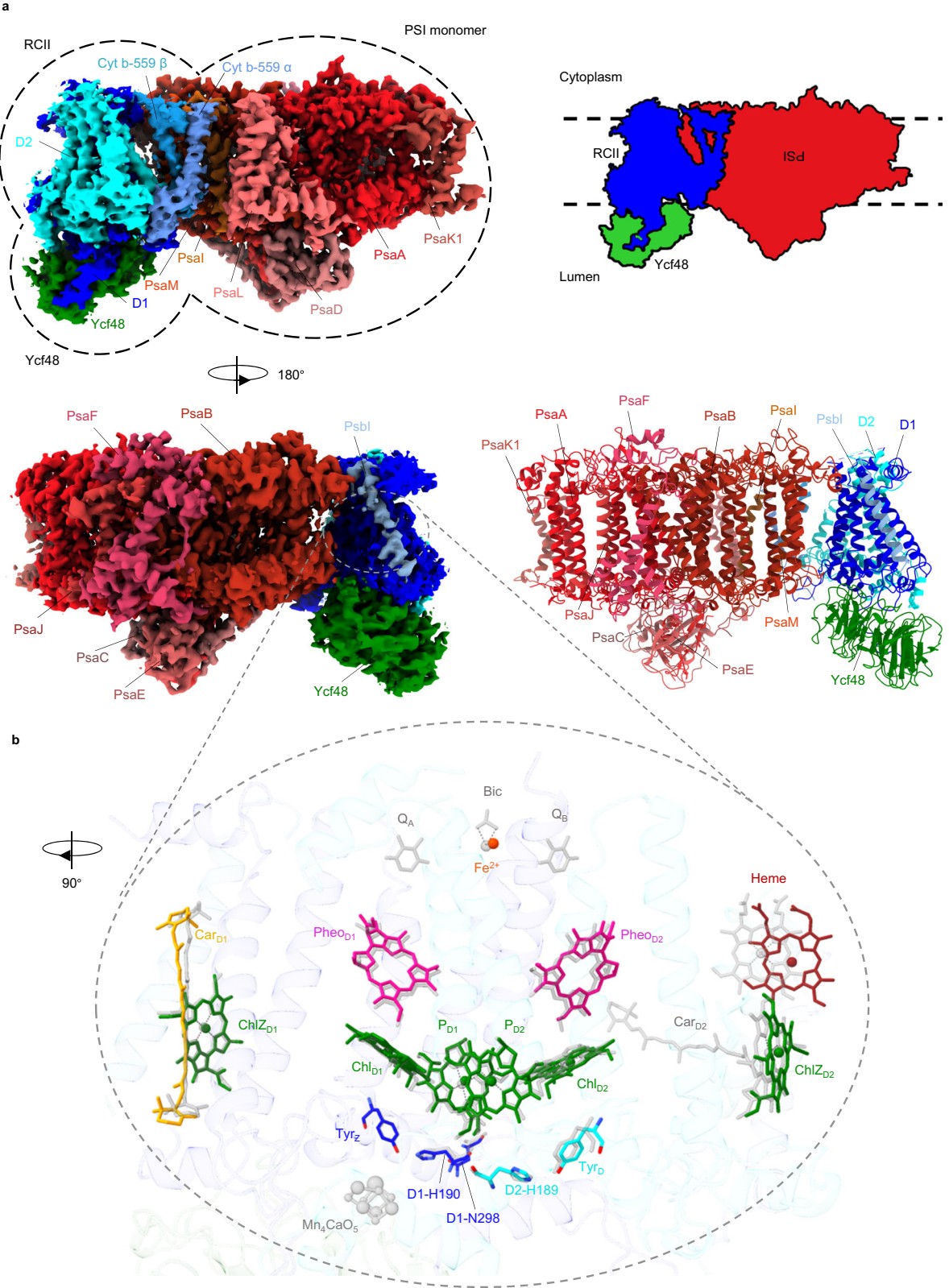

**Fig. 2 | Cryo-EM map of the RCII/PSI complex. a** The 11 PSI subunits in the PSI monomer, 5 PSII subunits in the RCII sub-complex and the Ycf48 assembly factor are colored and labelled. The PSI subunits are PsaA, PsaB, PsaC, PsaD, PsaE, PsaF, PsaI, PsaJ, PsaK1, PsaL and PsaM and the PSII subunits are D1, D2, PsbI, and the α (PsbE) and β (PsbF) subunits of Cyt b-559. **b** arrangement of co-factors and key amino-acid residues in RCII (in color) compared to the positions of the co-factors in oxygen-evolving Syn6803 PSII, shown in grey (PDB ID:7N8O)[25]. The phytol and isoprenoid chains of co-factors are hidden for clarity. Bic is a bicarbonate molecule, $Fe^{2+}$ is the non-heme iron, Chl represents chlorophyll, Car represents carotenoid, Pheo represents pheophytin, $Q_A$ and $Q_B$ are plastoquinones, $Tyr_Z$ is D1-Tyr161 and $Tyr_D$ is D2-Tyr160.

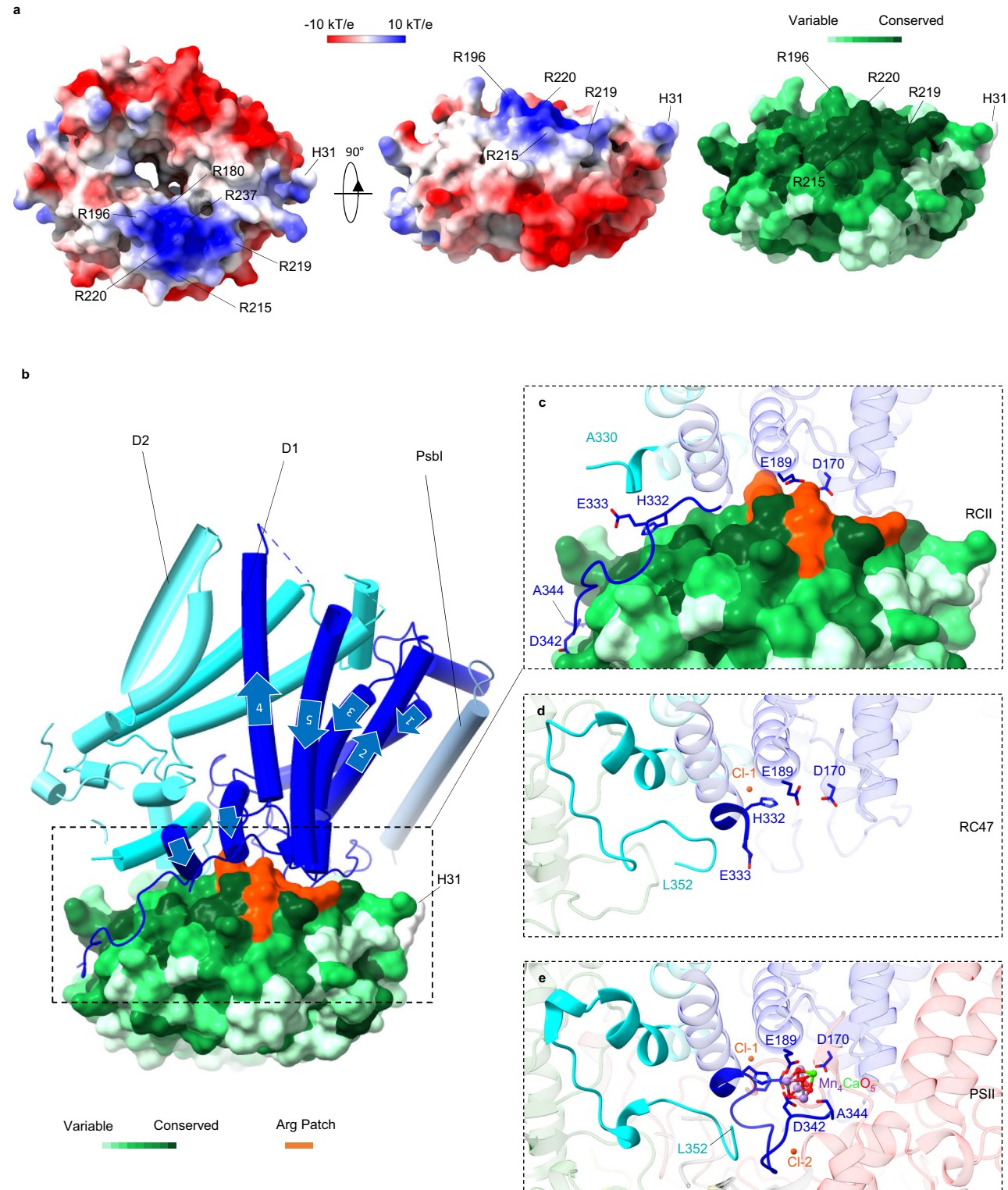

**Fig. 3 | Structure of Ycf48 and its binding site in the RCII complex. a** Top view looking from the thylakoid membrane towards the lumen (left panel) and side view (middle panel) of Syn6803 Ycf48 showing the electrostatic potential and the locations of the Arg patch and Ycf48-His31. A ConSurf analysis of Ycf48 (right panel) is coloured coded to indicate the degree of conservation of the residues. **b** Cartoon showing how D1 (dark blue) and D2 (cyan) bind Ycf48. The Arg patch is shown in orange and the direction of each of the five D1 transmembrane helices in the N-terminal to the C-terminal direction is indicated by arrows. **c** Interaction interface between D1/D2 and Ycf48 in RCII. The highlighted residues correspond to those that bind the Mn cluster in mature PSII. The final C-terminal 22 residues of D2 could not be resolved in the structure. **d** Location of the C-terminal regions of D1 and D2 in the RC47 complex (PDB ID:7DXA)[8]. The final C-terminal 11 residues of mature D1 could not be resolved. **e** The structure of the binding site of the $Mn_4CaO_5$ cluster in oxygen-evolving Syn6803 PSII (PDB ID:7N8O)[25]. Cl-1 and Cl-2 indicate the positions of the two chloride ions in PSII. CP47 shown in pale green and CP43 in pink. Panels (**c**, **d** and **e**) show D1 and D2 in the same orientation in the various structures.

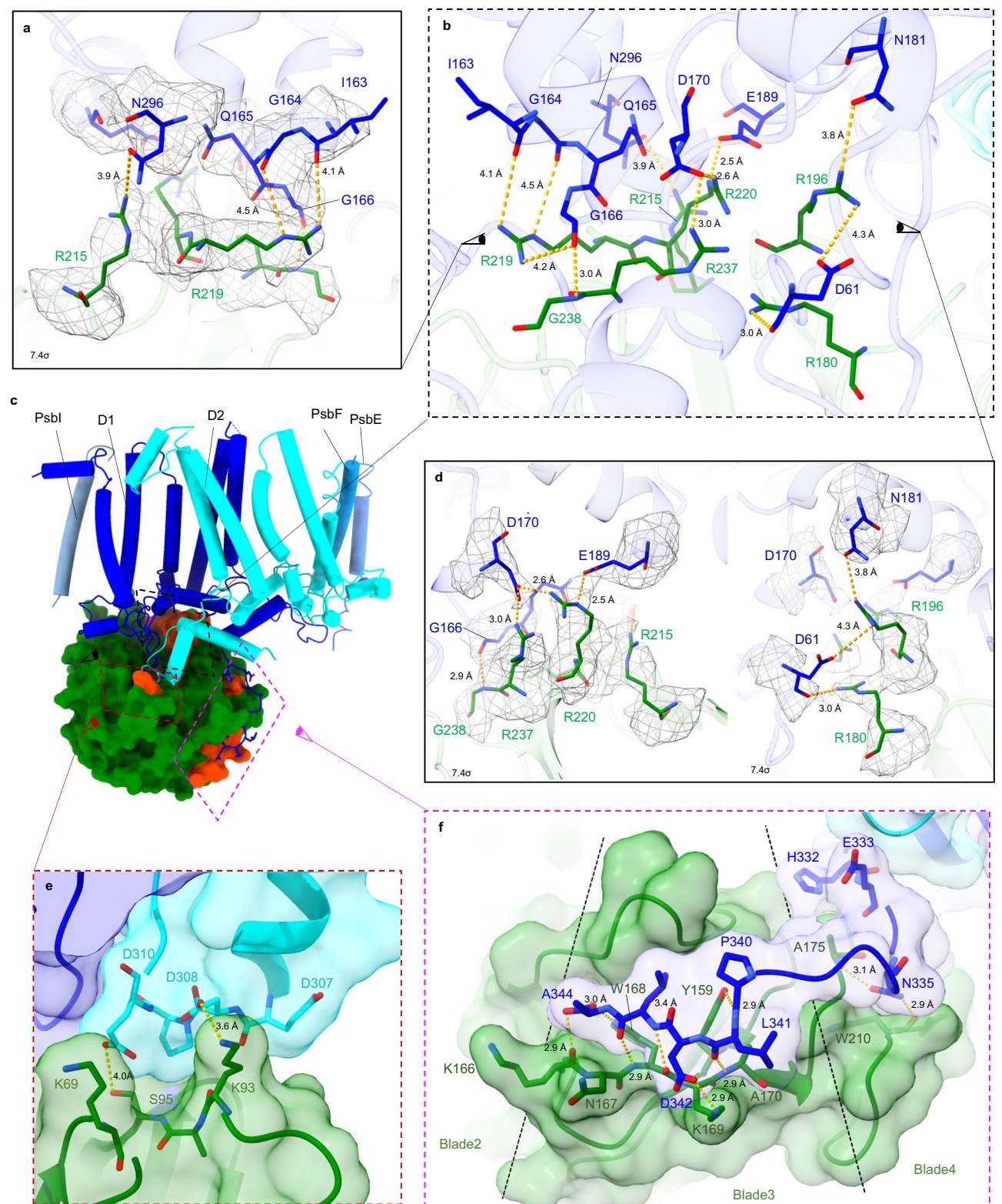

**Fig. 4 | Interaction of Ycf48 with D1 and D2. a**, **b**, **d** Interaction between Ycf48 and D1-Asp170 and D1-Glu189 showing map density at a contour level of 7.4σ, the fitted side chains and potential H-bonds viewed from different directions (the direction is indicated by sketched eyes in panels (**b** and **c**)). **c** Cartoon showing the location of the three regions of Ycf48, colored in orange, that interact with D1 or D2. **e** Interaction between D2 and Ycf48. **f** Interaction between the C-terminal tail of D1 and residues within Ycf48. Only close interactions (within 3.6 Å) are shown. The extent of blade 3 of Ycf48 is indicated by the two black dotted lines.

residues of mature D1 (Fig. 3c, Fig. 4c, f). Overall, the binding of Ycf48 to D1 disrupts the binding pocket of the $Mn_4CaO_5$ cluster in the D1 subunit and sterically prevents binding of Mn and other ions to D1 (Fig. 3c–e).

The D1 subunit is synthesised as a precursor with a C-terminal extension of 16 amino-acid residues which is processed to its mature form via an intermediate, iD1, carrying an 8 amino-acid extension[40]. The preparation subjected to cryo-EM contained approximately equal

amounts of both the mature and intermediate forms of D1 (Supplementary Fig. 3), but no amino-acid residues beyond D1-Ala344 were detected in the structure.

Previously we solved the crystal structure of *T. elongatus* Ycf48 in complex with an iD1 peptide (NAHNFPLDLASAESAPVA) containing the last 10 residues of mature D1 and the 8 amino-acid extension. The density of the entire peptide was observed in one of the non-crystallographic symmetry copies (PDB 5OJR chain E)[16]. In the RCII complex, the D1 peptide is in an equivalent position to the crystal structure between residues D1-Phe339 and D1-Ala344 (Fig. 4f). This previous work suggests that the D1 extension is capable of binding to Ycf48 so its absence in the cryo-EM structure may indicate that only complexes containing the mature form of the D1 subunit were imaged or that the mature D1 C-terminus is relatively homogeneous, whereas the position of the C-terminal extension is more heterogeneous and flexible.

Eukaryotic Ycf48 contains a characteristic 19 amino-acid insertion between blades 3 and 4 (PDB 5OJ3[16]). Superimposing the structure of Ycf48 from the red alga *Cyanidioschyzon merolae* onto Syn6803 Ycf48 suggests that the eukaryotic insertion is at the point where the D1 C-terminal tail leaves the membrane to interact with Ycf48, which may suggest a more extensive interaction between Ycf48 and D1 in eukaryotes (Supplementary Fig. 9). As eukaryotic Ycf48 lacks the N-terminal lipid anchor[35], the chloroplast protein may have acquired the extra loop to stabilize Ycf48 membrane-binding by other means.

Although Ycf48 interacts mainly with the lumenal surface of D1, Ycf48-Lys93 lies close to a negatively charged residue (D2-Asp308) in a loop in the C-terminal tail of D2 (Fig. 4e) that in mature PSII binds the PsbO subunit. This interaction between Ycf48 and D2 would help stabilise formation of the D1/D2 heterodimer during assembly of the RCII. Moreover, additional interactions of Ycf48 with CyanoP, an assembly factor bound to the lumenal surface of D2[41], but lost during purification, might also contribute to efficient RCII assembly.

An interesting feature of the Ycf48 beta-propeller structure is a central channel (Fig. 3a), which provides access to the region of D1 containing D1-Asp61 and D1-Glu65 with residue D1-Arg64 sitting at the exit (Supplementary Fig. 10). The narrowest diameter of the main central channel determined using MOLEonline is 4.4 Å (Supplementary Fig. 11). Bound $Na^+$ has been detected within the channel of the olfactomedin beta propeller domain[42] so the Ycf48 channel might plausibly act as a passage for $Ca^{2+}$, $Mn^{2+}$ and chloride ions.

## Structure of the PSI monomer

The structure of the PSI monomer is similar to an earlier structure of monomeric PSI from Syn6803 (PDB ID:6HQB)[43] except that the RCII/PSI complex contained PsaK1, rather than the PsaK2 isoform modelled previously, and 3 additional Chls and 4 additional beta-carotenes to give a total of 96 Chl and 24 carotenoids (Supplementary Fig. 12a). Two carotenoids were also shifted in position (Supplementary Fig. 12). Compared to the monomeric complex found in trimeric PSI (PDB ID:5OY0)[44], the PSI complex contains an additional Chl (PsaK1-1403 CLA; Supplementary Fig. 12b) in a similar location to a Chl found in tetrameric PSI from *Anabaena* sp. PCC 7120 (PDB ID: 6TCL)[45]. However, this chlorophyll is absent in two other structural models of the same complex (PDB ID:6K61[46] and PDB ID: 6JEO[47]).

## Discussion

We have discovered that the Ycf48 accessory factor binds to the lumenally exposed regions of D1 involved in binding the $Mn_4CaO_5$ oxygen-evolving cluster in the final holoenzyme. Our structural data therefore suggest that one role of Ycf48 is to prevent the premature binding and oxidation of Mn ions during PSII assembly so that the light-driven assembly of the cluster takes place at the appropriate stage of PSII biogenesis after attachment of CP47 and CP43. Current models suggest that the inappropriate assembly of the $Mn_4CaO_5$ cluster could lead to the formation of larger than normal Mn oxide clusters incapable of oxidizing water[48], defective clusters that produce reactive oxygen species[49] or the unwanted oxidation of other ions such as $Fe^{2+}$[50].

Previous work has suggested that the PratA protein of Syn6803 might pre-load unassembled D1 with $Mn^{2+}$ ions before D1 is incorporated into RCII[51]. However, there is no indication from the structure for the binding of Mn ions to the lumenal side of D1 in RCII. If PratA does have a role in delivering Mn to PSII, it would have to occur after detachment of Ycf48 or, alternatively, via a parallel assembly pathway lacking Ycf48, which is less plausible. Another candidate for delivering Mn to PSII is CyanoP, which is distantly related to the PsbP subunit[52] previously suggested to deliver Mn to PSII in chloroplasts[53,54]. However, whether CyanoP can bind Mn is still not clear.

Current models suggest that CyanoP binds to the lumenal side of unassembled D2 and aids formation of the RCIIa complex by interacting with Ycf48 attached to unassembled D1 (Fig. 1)[41]. Potentially, CyanoP might impact the binding of Ycf48 early in PSII assembly. However, neither CyanoP nor Ycf39 are required for binding of Ycf48 to RCIIa[18,41] and so loss of these accessory factors in the RCII/PSI complex is unlikely to have caused a major change in the mode of binding of Ycf48 to D1.

Ycf48 is a component of RCII assembly complexes but can be detected at low level in a sub-population of larger PSII complexes[16]. Such complexes might represent PSII assembly complexes or, possibly, repair complexes in which damaged D1 has been replaced by a newly synthesized D1 subunit, with Ycf48 still attached, as part of the PSII repair cycle[3].

An overlay of the structure of RC47[8] on the RCII complex reveals a clash between residues in the large lumenal domain of CP47 and residues in blade 3 of Ycf48 (Supplementary Fig. 13a) which might contribute to the detachment of Ycf48 from PSII or a change in its mode of binding. Comparison of RCII with the structure of a PSII assembly complex containing both CP47 and CP43, the latter attached to Psb27, reveals substantial clashes between the large lumenal loop in CP43 and Ycf48 (Supplementary Fig. 13b). Together, the binding of CP47 and the CP43/Psb27 complexes help expel Ycf48 from its binding site on D1 to form the Psb27-bound apo-PSII assembly complex (Supplementary Fig. 1). Subsequent light-driven assembly of the $Mn_4CaO_5$ cluster requires detachment of Psb27 and reorientation of the C-terminal tails of D1 and D2 (Fig. 3c–e)[7].

The D1/D2 structure on the acceptor side of RCII was rather ill-defined suggesting greater flexibility. This might in part be caused by loss of the RubA rubredoxin-like protein during purification of the RCII/PSI complex. RubA is speculated to bind close to the binding site of the non-heme iron in RCII, possibly to deliver the iron[17,55] or to play a photoprotective redox role[17]. Several other mechanisms might operate to protect the RCII assembly complex from photoinhibitory damage including energy spillover to the Ycf39/Hlips complex[18] or to PSI[12], cyclic electron flow around PSII[17] and, if quinone $Q_A$ is bound in vivo, an increase in the reduction potential of $Q_A$ to reduce the rate of singlet oxygen produced as a by-product of charge recombination[56].

Ycf48 co-purifies with the YidC insertase involved in the co-translational insertion of D1 into the membrane[16]. Current models suggest that Ycf48 might coordinate the packing of newly synthesized transmembrane helices of D1 with the insertion of chlorophyll and other co-factors[11,16]. The structure presented here is consistent with this suggestion as Ycf48 binds to the two loops and the C-terminal tail of D1 exposed to the lumen (Fig. 3b; Fig. 4b, c, f; Supplementary Fig. 10a). Given the critical importance of the C-terminal D1-Ala344 residue for assembly of the $Mn_4CaO_5$ cluster[57], binding of Ycf48 might also protect D1 from off-target proteolytic cleavage during biogenesis.

We found that the PSI monomer present in the RCII/PSI complex contained 3 additional Chl pigments in comparison with the PSI monomer described by[43] (PDB ID: 6HQB). However, our previous characterization of the PSI monomer in the RCII/PSI complex suggested considerable depletion of Chl after purification by Clear Native-PAGE[12]. Possible reasons for the discrepancy could be the release of some Chls during electrophoresis or association of the Chl-depleted PSI monomer with a sub-population of RCII complexes containing iD1 that might have escaped analysis by cryo-EM.

A peculiar feature of the RCII/PSI complex is the opposite membrane orientation of RCII and PSI in comparison with the standard PSI and PSII complexes embedded in the thylakoid membrane. Neither standard negative staining EM nor cryo-EM could detect any RCII/PSI complexes with the expected "standard" orientation. Our current understanding of the biogenesis of photosynthetic complexes would suggest that this unusual PSI/PSII orientation in the complex is artefactual. Regardless of its origin, the formation of a stable RCII/PSI complex has allowed us to determine the structural details of Ycf48 attachment to an RCII early assembly complex. The interaction interface between PSI and RCII does not overlap with the binding site for Ycf48 (Fig. 2), further supporting the validity of the findings regarding the interaction between Ycf48 and D1/D2.

Great progress has been made recently in determining the structures of various precursor complexes involved in the assembly, and, possibly, the repair of PSII. The Psb27 accessory factor binds to the CP43 subunit and appears to indirectly prevent assembly of the mature PSII complex by sterically preventing binding of the PsbO extrinsic protein[6,7]. Psb27-containing complexes might act as pool of PSII complexes that can be rapidly activated to maintain PSII homeostasis[58]. Psb28 binds to the D1, D2 and CP47 subunits on the cytoplasmic side of the RC47 complex and the non-oxygen-evolving PSII monomeric complex and induces large-scale conformational changes around the non-heme iron and $Q_B$-binding site, probably as a photoprotective mechanism[7,8]. Here we provide structural information on an early assembly RCII complex and show that Ycf48 binds to the key residues in D1 involved in ligating the oxygen-evolving $Mn_4CaO_5$ cluster and must be detached to allow the necessary remodelling of the C-terminal regions of D1 and D2 to form the binding site for the cluster.

## Methods

### Isolation and analysis of the RCII/PSI complex

His-tagged RCII complexes were isolated from a *Synechocystis* strain expressing a His-tagged version of D2 and lacking CP47 (strain His-D2/ΔCP47) which was constructed and cultivated as described in[12,35] and thylakoid membranes were obtained as in[59]. The preparation was purified using Protino Ni-NTA agarose (MACHEREY-NAGEL, Germany) in a gravity-flow chromatography column at 10 °C after membrane solubilization with 1.5% (w/v) n-dodecyl-β-D-maltoside (DM) as described in[35], with the exception that buffer A lacking glycerol was used for all purification steps. The final eluate was concentrated about 30-fold using an Amicon Ultracel 100 K. The final volume of 100 μl was then loaded onto a continuous 10–30% (w/v) sucrose gradient and the RCII complexes were separated by ultracentrifugation (40,000 × g, 15 h) using an Optima XPN-90 Ultracentrifuge (Beckman Coulter, USA). The separated RCII/PSI fraction was collected and then washed repeatedly using the glycerol- and DM-free elution buffer and concentrated to 0.2 mg Chl/ml which corresponded to approximately 0.6 mg protein/ml.

The protein composition of complexes was analyzed by clear native (CN) electrophoresis in a 4% to 14% gradient polyacrylamide gel or by SDS-PAGE in a denaturing 16% to 20% gradient gel containing 7 M urea as in[58] (Supplementary Figs. 2, 3). The uncropped data have been provided at the end of supplementary files. The absorption spectrum

was measured using a Shimadzu UV3000 spectrophotometer and 77 K Chl fluorescence spectrum using a P.S.I. fluorimeter with excitation light at 470 nm.

### Grid preparation

Quantifoil 0.6/1 300 mesh copper grids were manually coated with an approximately 1 nm layer of continuous carbon and glow discharged just before usage for 5 s at about 25 mA. 3 μl of fresh sample (at 0.2 mg protein/ml, or 0.06 mg chlorophyll/ml), containing 0.1% FOM (Fluorinated Octyl Maltoside) to reduce protein denaturation at the air-water interface[60], were applied to the grids in a humidified chamber (100% humidity at 4 °C) of a Vitrobot Mark IV, blotted for 2 s with 25 force and plunge-frozen in liquid ethane.

### Data collection

The dataset was collected on a TFS Glacios microscope, equipped with a FalconIII camera. Micrographs were acquired in linear mode, at a nominal pixel size of 1.2 Å. The total dose applied to the micrographs was 91 e-/Å$^2$, with 58 frames per micrograph and a defocus range of −1.2 μm to −2.5 μm. After centring the beam in a hole and collecting a micrograph, the surrounding 8 holes were imaged using beam-image-shift[61] to improve the throughput. The details about the data collection are listed in Supplementary Table 2.

### Data processing

The processing pipeline is illustrated in Supplementary figure 4. The micrographs were mainly processed using Relion/3.1[19]. MotionCor2[62] was used for motion correction, CTFFIND4[63] for the initial estimation of CTF parameters, gctf[64] for per-particle defocus estimation and Topaz[20] for accurate picking after preliminary processing. Local resolution was calculated using ResMap 1.1.4. The maps were refined both as overall consensus refinement for each class and by focusing on PSI using a pool of particles from the two classes, leading to the following gold-standard resolutions: 3.2 Å for class 2; 3.1 Å for class 3; 2.9 Å for pooled focused refinement on PSI.

As outlined in Supplementary Fig. 4a, after motion correction and CTF estimation, particles were picked with LoG (Laplacian of Gaussian) spatial filter in Relion and subjected to 2D and 3D classification, to obtain good references for Topaz picking. After Topaz picking with the best classes, per-particle defoci were estimated and two more rounds of 3D classification separated the particles of interest from remaining junk and contamination. Subsequently, 3D refinement, CTF-refinement and polishing were applied to the pool of good particles, which were then classified without alignment to separate the two different conformations. These maps were then globally refined (two classes separately) and focus-refined around PSI (using the pooled classes). Focused-refinement of PSII with the pooled classes did not give better results than the global refinements.

### Model fitting

The structures of the monomer of PSI (PDB ID: 5OY0[44]) [https://www.rcsb.org/structure/5OY0], the D1-D2-PsbI-PsbE-PsbF portion of apo-PSII (PDB ID: 6WJ6[25]) [https://www.rcsb.org/structure/6WJ6], and Ycf48 (PDB 2XBG[16]) [https://www.rcsb.org/structure/2XBG] were placed with phenix.dock_in_map then rebuilt in COOT with cycles of refinement in phenix.real_space_refine[65]. The structure was validated with molprobity[66]. Refinement statistics are shown in Supplementary Table 2. Structure figures were made with UCSF ChimeraX[67] and the ConSurf analysis of Ycf48 was done according to[16] using the ConSurf server at https://consurf.tau.ac.il/ [68,69]. Briefly, 382 unique homologs of Syn6803 Ycf48 were collected from UNIREF90 (Uniport) by HMMER with an E-value of 0.0001 or less and a sequence identity between 35% and 95%. 150 sequences (listed in Supplementary Table 3), which is the maximum number that can be analyzed, were aligned by MAFFT, and a conservation score for each residue was assigned from 1 with the

lowest score to 9 with the highest score. The residues of Ycf48 were coloured according to their conservation score, from light green (score 1) to deep green (score 9) in the main figures and from turquoise (score 1) to maroon (score 9) in the supplementary figures. Amino-acid positions for which the inferred conservation level was assigned with low confidence were coloured yellow.

The electrostatic potential shown in Fig. 3a was generated in UCSF ChimeraX, with the default blue (positive) to red (negative) color scheme applied with a dielectric constant of 4 (https://www.cgl.ucsf.edu/chimerax/docs/user/commands/coulombic.html). Structural models used for comparison and their PDB ID in brackets are: Syn6803 PSI trimer (PDB ID:5OY0)[44] [https://www.rcsb.org/structure/5OY0], Syn6803 oxygen-evolving PSII (PDB ID: 7N8O)[25] [https://www.rcsb.org/structure/7N8O], *Thermosynechococcus vulcanus* RC47 (PDB ID: 7DXA)[8] [https://www.rcsb.org/structure/7DXA], *Thermosynechococcus vulcanus* Psb27/PSII dimer (PDB ID: 7CZL)[6] [https://www.rcsb.org/structure/7CZL], Syn6803 PSI monomer (PDB ID: 6HQB)[43] [https://www.rcsb.org/structure/6HQB], *Cyanidioschyzon merolae* Ycf48 (PDB ID: 5OJ3)[16] [https://www.rcsb.org/structure/5OJ3]. *Thermosynechococcus elongatus* Ycf48 (PDB ID: 2XBG)[16] [https://www.rcsb.org/structure/2XBG] was used as an initial model to determine the structure of Syn6803 Ycf48. The UniProt database accession codes for the various PSI and PSII proteins can be found in Supplementary Table 1.

### Reporting summary
Further information on research design is available in the Nature Portfolio Reporting Summary linked to this article.

## Data availability
The cryo-EM density maps are deposited in the Electron Microscopy Data Bank under accession numbers EMD-15618 (class 2, 3.2 Å), EMD-15522 (class 3, 3.1 Å) and EMD-15621 (PSI focused, 2.9 Å). The atomic models of the cryo-EM structures are deposited in the Protein Data Bank under accession numbers 8ASL (class 2, 3.2 Å), 8AM5 (class 3, 3.1 Å) and 8ASP (PSI focused, 2.9 Å).

The following source data were used in this paper: the D1-D2-PsbI-PsbE-PsbF portion of apo-PSII (PDB ID: 6WJ6[25]) [https://www.rcsb.org/structure/6WJ6], *Thermosynechococcus elongatus* Ycf48 (PDB 2XBG[16]) [https://www.rcsb.org/structure/2XBG], Syn6803 PSI trimer (PDB ID:5OY0)[44] [https://www.rcsb.org/structure/5OY0], Syn6803 oxygen-evolving PSII (PDB ID: 7N8O)[25] [https://www.rcsb.org/structure/7N8O], *Thermosynechococcus vulcanus* RC47 (PDB ID: 7DXA)[8] [https://www.rcsb.org/structure/7DXA], *Thermosynechococcus vulcanus* Psb27/PSII dimer (PDB ID: 7CZL)[6] [https://www.rcsb.org/structure/7CZL], Syn6803 PSI monomer (PDB ID: 6HQB)[43] [https://www.rcsb.org/structure/6HQB], *Cyanidioschyzon merolae* Ycf48 (PDB ID: 5OJ3)[16] [https://www.rcsb.org/structure/5OJ3], *Anabaena* sp. PCC 7120 PSI (PDB ID: 6TCL)[45], *Anabaena* sp. PCC 7120 PSI (PDB ID:6K61)[46] [https://www.rcsb.org/structure/6K61], and *Anabaena* sp. PCC 7120 PSI (PDB ID: 6JEO)[47] [https://www.rcsb.org/structure/6JEO].

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

## Acknowledgements
P.J.N. and J.W.M. are grateful for the support of the Biotechnology & Biological Sciences Research Council (awards BB/L003260/1 and BB/P00931X/1). J. Knoppová, R.S. and J. Komenda were supported by the Czech Science Foundation (project 19-29225X) and by ERC project Photoredesign (no. 854126) and L.A.S. was supported by the Scientific Service Units (SSU) of IST Austria through resources provided by the Electron Microscopy Facility (EMF), the Life Science Facility (LSF) and the IST high-performance computing cluster.

## Author contributions
P.J.N., L.A.S., and J. Komenda conceived the research; I.V., J. Knoppová and R.S. performed the experiments; Z.Z., I.V. and J.W.M. analyzed the data; all authors contributed to writing of the manuscript.

## Competing interests
The authors declare no competing interests.
