## [Peer Review File · Nature Communications]

The Ycf48 accessory factor occupies the site of the oxygen-evolving manganese cluster during photosystem II biogenesisREVIEWER COMMENTS

Reviewer #1 (Remarks to the Author):

This manuscript reports on the structural characterization of an early PSII assembly intermediate from the cyanobacterium *Synechocystis* spp. PCC6803 containing the PSII assembly factor Ycf48. Previous genetic and biochemical work of the authors had revealed the functional role of Ycf48 during the early PSII assembly steps leading to the formation of RCII complexes. Moreover, structural analysis of Ycf48 complexed with a peptide of the D1 C-terminus had revealed the 7-bladed beta propeller structure of this factor by the same groups.

Here, the authors provide structural evidence for the binding mode of Ycf48 within an RCII-like complex preparation containing the PSII subunits D1, D2, PsbI, PsbE and PsbF as well as a most likely artificially attached PSI moiety. Other RCII-associated assembly factors including CyanoP, Ycf39, RubA and HliD were not resolved in this structure. The data reveal a precise view on the interaction of Ycf48 to the luminal tail of D1 where the Mn cluster is ligated in mature PSII. In addition, the authors identify very interesting tunnels in Ycf48 that might serve in ion transport to D1 although no bound ions were detected. Therefore, this work provides new important insights into the early PSII assembly steps which have not been structurally resolved before. Nevertheless, this reviewer has some questions and comments as outlined below.

Most likely (as stated also by the authors), the RCIIa/PSI complex is artificially formed as indicated by the inverse orientation of the PSI complex part. On the one hand, this increased the size of the complex and thereby facilitated its analysis, however, important other assembly factors are lost during purification or possibly detached by PSI from the RCII complex, i.e., CyanoP, RubA, Ycf39 and HliD. In particular, the lack of CyanoP might significantly affect Ycf48 binding to D1 (in Fig. 1, CyanoP should be omitted from the scheme of the RCIIa/PSI complex). To this reviewer, this should be discussed a bit more thoroughly also with regard to Mn transport into PSII because the plant relative of CyanoP, namely PsbP, has been suggested to bind Mn ions. Likely, the smaller RCII complexes (RCIIa and RCII*) represent more native states during PSII assembly. Despite their relatively small size, have the authors tried to identify them via cryo EM?

With regard to the presence of the mentioned assembly factors, Suppl. Tab. 1 is not fully clear to me. Their abundance was determined via Western blotting but what does +/- reflect? For instance, Ycf39 appears to be present in stoichiometric amounts based on the stained gel (suppl. Fig. 3). It might be informative to present also the Western analysis. Have the authors tested for the presence of CyanoP, too?

One puzzling finding is that the iD1 processing intermediate is also present in stoichiometric amounts in the RCIIa/PSI fraction. But, it is not resolved in the structure despite its capability to bind to Ycf48 based on the crystal Ycf48 structure (lanes 210-212). The authors should provide an idea why they apparently imaged only complexes containing mature D1.

Reviewer #2 (Remarks to the Author):

This manuscript describes the structure of a cyanobacterial PSII D1/D2 reaction center assembly complex with Ycf48 attached, solved by cryo-electron microscopy. This is an early PSII-assembly intermediate, and the binding of Ycf48 appeared to prevent the assembly of the Mn₄CaO₅ cluster that catalyzes water oxidation. Overall, this is a novel and interesting contribution, and I can recommend the acceptance of this manuscript provided that the authors take into consideration the following points in revising their manuscript.

In Supplementary Fig. 1, the CP47 module alone lacks Psb34, but it is appeared in the RC47 complex. Is there any evidence that Psb34 is incorporated into the RC47 complex during joining of the CP47 module with the RCII* complex? If not, it looks more natural by putting Psb34 into the CP47 module as it is a trans-membrane subunit and bound directly to the CP47 subunit.

Legend to Supplementary Fig. 2, it was written that “The upper band in the sucrose gradient contains the (Ni+SGupper band) RCII/PSI...”. It seems to me that the “upper band” and “lower band” are inverted here. Please check.

Supplementary Fig. 3, in RCII/PSI, what are the bands below PsaA/B, below D1, and the band above PsaD? These three bands look quite dense, but are not labeled.

Legend to Fig. 2, “...the PSII subunits are D1...”. Here only 5 PSII subunits are listed, but no Ycf48. Where Ycf48 should be listed?

Lines 186-188: “...potential weak hydrogen bonds (3.8 Å– 4.5 Å) are present between ...

and between Ycf48-Arg219 and the backbone (carbonyl groups) of D1-Ile163, D1-Gly164 and D1-Gly166 (Fig. 4a).” In Fig. 4a, I cannot see the bonds between Ycf48-Arg219 and D1-Gly166. Where is it and what is the distance? It is also better to label all the distances in Fig. 4b to aid understanding of the readers.

Legend to Supplementary Fig. 10, please add a reference after “IJMS analysis”.

Lines 236-238: “...the PSI complex contains an additional Chl (Supplementary Fig. 12) in a similar location to a Chl found in tetrameric PSI from *Nostoc* sp. PCC 7120”. Is this Chl also present in PSI tetramer from *Anabaena* in two other reported structures (Zheng L. et al. *Nat Plants*. 2019; Kato K, et al. *Nat. Commun*. 2019)? This should be made clear and these two refs. should be cited here.

Discussion part: It was described that “one role of Ycf48 is to regulate the binding and oxidation of Mn ions during PSII assembly”. However, it is known that Psb27 is bound to apo-PSII that lacks the Mn₄CaO₅ cluster, and the assembly of the Mn₄CaO₅ cluster occurs after detachment of Psb27. In Psb27-bound apo-PSII, Ycf48 is not bound, suggesting that the immediate precursor for the Mn₄CaO₅ cluster assembly is the Psb27-bound PSII. Thus, Ycf48 seems to aid in the assembly of Psb27-bound apo-PSII, instead of the mature PSII. On the other hand, the crashes of mature CP47 and CP43 with Ycf48 seems to indicate the necessity of Ycf48 in preventing the unwanted folding and assembly of CP47 and CP43 before they are matured and incorporated into PSII correctly. This needs to be discussed explicitly.

Line 332: What is FOM?

Reviewer #3 (Remarks to the Author):

Photosystem II (PSII) found in photosynthetic organisms is fundamental in defining earth’s biosphere. It acts as the gateway for energy from the sun and it releases oxygen that allows for all higher life on our planet. The study of PSII is widely applicable due to its relevance in fields ranging from synthetic solar fuel catalysis to metalloenzyme assembly. A major gap in our understanding of PSII is its biogenesis, which is vitally important because PSII turnover in vivo is rapid which confers resilience to oxidative damage to the complex. Only recently have some assembly intermediates found during PSII biogenesis been thoroughly characterized, primarily due to new structural data that have provided bases for biochemical interpretation. These assembly intermediates have revealed important features of PSII biogenesis, but primarily those that occur in late-stage assembly. A thorough understanding of early-stage biogenesis intermediates have so far remained elusive. This represents a gap in our understanding of this key enzyme.

Here, Zhao et al. reveal the structural basis of an early biogenesis intermediate of PSII. In the PSII holocomplex, water oxidation is catalyzed by a tetramanganese cluster in the active site. In the assembly intermediate described here, the active site is blocked by an assembly factor called Ycf48 which most

likely prevents the premature binding of ions that eventually comprise the metallofactor. Based on the conservation observed in the regions interacting with the active site, it is likely that this blocking strategy is relevant to all oxygenic phototrophs. This novel insight enhances our understanding of PSII biogenesis, and it is communicated extremely well both in writing and in figures. I strongly recommend the manuscript for publication in Nature Communications pending some minor edits that are listed here:

PSI structure: I cannot tell how the authors have modeled P700 because I do not have the coordinates, but please use two different residues, CLA and CLO. The CLO is the Chl a prime that has reversed stereochemistry about its 13-2 carbon atom. 5OY0 which the authors have used as a starting model has this incorrectly modeled by two standard Chl a molecules. An example of its proper use is in PDB 6NWA.

Supp Fig. 5: Please add labels to the figure. It would be helpful to designate which parts correspond to RCII and PSI, and the view perspective, such as “stromal side view” or “membrane plane view”.

Line 155: I suggest commenting a bit further on the Cl⁻ ions. Are those regions instead filled with something else? For Cl-2, I imagine this space is instead occupied by Ycf48. Are there any features occupying the space of Cl-1? I also suggest pointing out that although no Cl⁻ ions were identified in the map, anions are especially challenging to resolve in cryo-EM maps (as opposed to X-ray crystallography-derived electron density maps) unless the resolution is very high. This is due to the negative electrostatic potential scattering factors of anions. An alteration to the text such as “Importantly, there was no indication in the structure for the presence of bound Mn or Ca ions. Additionally, the two chloride ions, called Cl-1 and Cl-2, were not identified in the cryo-EM map, although it should be noted that anions are challenging to resolve at medium resolution in cryo-EM (REF)” would be sufficient. An appropriate citation would be Wang 2017 (<https://doi.org/10.1002/pro.3198>). Depending on how the coauthors edit this, they may also want to revise the sentence starting on line 262.

Line 174/Fig.3/Supp Fig. 10: Please describe how conservation was determined and cite ConSurf, or at least include a link to it. Were the authors required to input sequences? Were they aligned? If so, how? The authors could add this as a small paragraph in the methods section.

Supp. Fig. 10: There is an incomplete reference. Also, the last part of the sentence doesn't make sense to me. What are the authors trying to describe about ConSurf here?

Line 208: Typo – the authors meant to close the parentheses instead of including a semicolon.

Line 210: I suggest that the authors clarify this sentence. On line 90, the authors stated “Both precursor and mature forms of D1 were detected in the RCII/PSI preparation suggesting a heterogeneous

population of complexes.” Therefore, I don’t think that only mature D1 forms were imaged during data collection, which is how I read it. Perhaps they mean that the ensemble 3D reconstruction (i.e., the cryo-EM map) only captures the fraction of particles that contain the mature form of D1? The authors may suggest that the mature D1 C-terminus is relatively homogeneous in its position, but that the unprocessed D1 C-terminal extension positions are heterogeneous.

Line 216: Does the red algal Ycf48 homolog have a lipid anchor? Note the inverse correlation between binding affinity and lipid anchor discussed for PsbQ and the Psb27 assembly factor reported recently <https://doi.org/10.1007/s11120-021-00888-2>. The authors may want to add a comment accordingly.

REVIEWER COMMENTS

Reviewer #1 (Remarks to the Author):

Most likely (as stated also by the authors), the RCIIa/PSI complex is artificially formed as indicated by the inverse orientation of the PSI complex part. On the one hand, this increased the size of the complex and thereby facilitated its analysis, however, important other assembly factors are lost during purification or possibly detached by PSI from the RCII complex, i.e., CyanoP, RubA, Ycf39 and HliD. In particular, the lack of CyanoP might significantly affect Ycf48 binding to D1

Reply: Our previous work has provided evidence that CyanoP binds to unassembled D2, not unassembled D1, and facilitates the formation of RCII via its interaction with Ycf48 attached to unassembled D1 (Knoppová et al., 2016). Our earlier studies have also suggested that CyanoP is present mainly in the RCII* complex which also contains the Ycf39/HliP complex (Knoppová et al., 2016; Knoppova et al., 2022).

However, *in vivo*, absence of either Ycf39 or CyanoP does not markedly affect formation of the RCIIa complex containing bound Ycf48 and the phenotype of both null mutants is similar to that of wild type (Knoppová et al., 2014; Knoppová et al., 2016), whereas the lack of Ycf48 seriously obstructs the assembly of RCII (Komenda et al., 2008). These data therefore strongly suggest that CyanoP and Ycf39 do not have a major effect on the binding of Ycf48 to D1 and that consequently the mode of attachment of Ycf48 to D1 in the RCII/PSI complex is similar to that found *in vivo*. Our structural data in the submitted manuscript are also in line with our previous mutagenesis work indicating an important role for the 'Arg patch' in binding to PSII *in vivo* (Yu et al., 2018).

However, we do recognise that loss of CyanoP and Ycf39 could potentially cause some minor changes in the mode of Ycf48 binding and agree with the reviewer that this should be mentioned.

Author action: We have now added the statement 'Current models suggest that CyanoP binds to the luminal side of unassembled D2 and aids formation of the RCIIa complex by interacting with Ycf48 attached to unassembled D1 (Fig. 1) (Knoppová et al., 2016). Potentially, CyanoP might impact the binding of Ycf48 early in PSII assembly. However, neither CyanoP nor Ycf39 are required for binding of Ycf48 to RCIIa (Knoppová et al., 2014; Knoppová et al., 2016) and so loss of these accessory factors in the RCII/PSI complex is unlikely to have caused a major change in the mode of binding of Ycf48 to D1.'

(in Fig. 1, CyanoP should be omitted from the scheme of the RCIIa/PSI complex).

Author action: As suggested by the reviewer we have removed CyanoP from the RCIIa/PSI complex in Fig. 1.

To this reviewer, this should be discussed a bit more thoroughly also with regard to Mn transport into PSII because the plant relative of CyanoP, namely PsbP, has been suggested to bind Mn ions. Likely, the smaller RCII complexes (RCIIa and RCII) represent more native states during PSII assembly.*

The reviewer makes a valuable point about CyanoP and PsbP and their possible role in delivering Mn to PSII. Recent results using X-ray crystallography have provided important evidence to suggest that the PsbP can bind manganese (Cao et al., 2015). If this feature holds for the homologous CyanoP protein, then CyanoP might help increase the local concentration of Mn ions during photoactivation of PSII during biogenesis. However, there is still no evidence that CyanoP binds Mn (although the binding of Zn has been detected in crystals; Michoux et al. (2010)) and although CyanoP is homologous to PsbP, its closest homologue in chloroplasts is the PLP1 protein which has been shown to be involved in PSII assembly, but which has not yet been shown to bind Mn.

Author action: To summarise these points, we include in the text ‘Another candidate for delivering Mn to PSII is CyanoP, which is distantly related to the PsbP subunit (Michoux et al., 2010) previously suggested to deliver Mn to PSII in chloroplasts (Bondarava, Un & Krieger-Liszskay, 2007; Cao et al., 2015). However, whether CyanoP can bind Mn is still not clear.’

Despite their relatively small size, have the authors tried to identify them via cryo EM?
We agree that smaller RCII* and RCIIa complexes might provide useful information in future studies, but from our experience this approach is likely to be relatively challenging due to their small size.

With regard to the presence of the mentioned assembly factors, Suppl. Tab. 1 is not fully clear to me. Their abundance was determined via Western blotting but what does +/- reflect? For instance, Ycf39 appears to be present in stoichiometric amounts based on the stained gel (suppl. Fig. 3). It might be informative to present also the Western analysis. Have the authors tested for the presence of CyanoP, too?

Author action: In light of the reviewer’s comments, we have modified the designation + and +/- to ++ and + in Supplementary Table 1 to designate the abundance of the protein in the preparation, based on the intensity of the staining on SDS-PAGE gels and signals in immunoblotting experiments, now included in a revised version of Supplementary Fig. 3. Ycf39, RubA and HliD were all detected in the RCII preparation but were depleted in the final RCII/PSI preparation when compared to the major components (Ycf48, D1 and D2). The reason that Ycf39 looks more abundant in the stained gel is that the sharp band above D2 assigned to Ycf39 overlaps with a more diffuse band of the contaminant delta-aminolevulinic dehydratase (P77969). CyanoP was not detected in the RCII/PSI preparation which is in agreement with previous analyses (Knoppová et al., 2022).

One puzzling finding is that the iD1 processing intermediate is also present in stoichiometric amounts in the RCIIa/PSI fraction. But, it is not resolved in the structure despite its capability to bind to Ycf48 based on the crystal Ycf48 structure (lanes 210-212). The authors should provide an idea why they apparently imaged only complexes containing mature D1.

The ratio iD1/D1 in RCII/PSI is close to one and, as correctly stated by the reviewer, we were only able to model mature D1 in the structure. The reason is not yet clear, but we suggest two possibilities in the revised manuscript: that the iD1-containing RCII/PSI complexes do not interact well with the carbon grid and are not imaged or the C-terminal extension is too flexible to be detectable by cryo-EM at high resolution.

Reviewer #2 (Remarks to the Author):

This manuscript describes the structure of a cyanobacterial PSII D1/D2 reaction center assembly complex with Ycf48 attached, solved by cryo-electron microscopy. This is an early PSII-assembly intermediate, and the binding of Ycf48 appeared to prevent the assembly of the Mn₄CaO₅ cluster that catalyzes water oxidation. Overall, this is a novel and interesting contribution, and I can recommend the acceptance of this manuscript provided that the authors take into consideration the following points in revising their manuscript.

In Supplementary Fig. 1, the CP47 module alone lacks Psb34, but it is appeared in the RC47 complex. Is there any evidence that Psb34 is incorporated into the RC47 complex during joining of the CP47 module with the RCII complex? If not, it looks more natural by putting Psb34 into the CP47 module as it is a trans-membrane subunit and bound directly to the CP47 subunit.*

We would first like to thank the reviewer for their positive comments about the novelty of our work.

The reviewer makes a good point about the role of Psb34 in assembly and when it binds to CP47. It is still not clear when Psb34 binds during PSII assembly, but recent models by Komenda and co-workers (Rahimzadeh-Karvansara et al., 2022) propose a role for Psb34 in displacing the Hlip complex from CP47 under high light conditions.

Author action: Given this, we propose to keep Supplementary Fig. 1 as it is, but now include in the legend that there is still uncertainty on this point.

Legend to Supplementary Fig. 2, it was written that “The upper band in the sucrose gradient contains the (Ni+SGupper band) RCII/PSI...”. It seems to me that the “upper band” and “lower band” are inverted here. Please check.

Author action: We thank the reviewer for spotting the mistake, which we have now corrected in the revised manuscript.

Supplementary Fig. 3, in RCII/PSI, what are the bands below PsaA/B, below D1, and the band above PsaD? These three bands look quite dense, but are not labeled.

The upper band is elongation factor EF-Tu, which is a common contaminant in this type of preparation (see Knoppová et al. 2022). This band is now labelled. Unfortunately, we have not yet succeeded in identifying the other two bands by mass spectrometry, despite their relative high abundance.

Legend to Fig. 2, “...the PSII subunits are D1...”. Here only 5 PSII subunits are listed, but no Ycf48. Where Ycf48 should be listed?

Ycf48 is listed as “assembly factor” in the legend.

Lines 186-188: “...potential weak hydrogen bonds (3.8 Å– 4.5 Å) are present between ... and between Ycf48-Arg219 and the backbone (carbonyl groups) of D1-Ile163, D1-Gly164 and D1-Gly166 (Fig. 4a).” In Fig. 4a, I cannot see the bonds between Ycf48-Arg219 and D1-Gly166. Where is it and what is the distance? It is also better to label all the distances in Fig. 4b to aid understanding of the readers.

Author action: As requested by the reviewer, we have now modified Fig. 4 so that the H-bond is shown between Ycf48-Arg219 and D1-Gly166 in Fig 4b and have labelled all the distances in Fig. 4b.

Legend to Supplementary Fig. 10, please add a reference after “IJMS analysis”.

Author action: We have reworded this sentence so that there is now no need to cite the reference in the International Journal of Molecular Science.

Lines 236-238: “...the PSI complex contains an additional Chl (Supplementary Fig. 12) in a similar location to a Chl found in tetrameric PSI from Nostoc sp. PCC 7120”. Is this Chl also present in PSI tetramer from Anabaena in two other reported structures (Zheng L. et al. Nat Plants. 2019; Kato K, et al. Nat. Commun. 2019)? This should be made clear and these two refs. should be cited here.

Author action: As suggested by the reviewer, we have now added text to make clear that the additional chlorophyll is only present in the tetramer structure reported by Chen et al (2020).

Added text: Compared to the monomeric complex found in trimeric PSI (PDB ID:5OY0) (Malavath et al., 2018), the PSI complex contains an additional Chl (PsaK1-1403 CLA; Supplementary Fig. 12b) in a similar location to a Chl found in tetrameric PSI from *Anabaena* sp. PCC 7120 (PDB ID: 6TCL). However, this chlorophyll is absent in two other structural models of the same complex (PDB ID:6K61 (Zheng et al., 2019) and PDB ID: 6JEO (Kato et al., 2019).

Discussion part: It was described that “one role of Ycf48 is to regulate the binding and oxidation of Mn ions during PSII assembly”. However, it is known that Psb27 is bound to apo-PSII that lacks the Mn₄CaO₅ cluster, and the assembly of the Mn₄CaO₅ cluster occurs after detachment of Psb27. In Psb27-bound apo-PSII, Ycf48 is not bound, suggesting that the immediate precursor for the Mn₄CaO₅ cluster assembly is the Psb27-bound PSII. Thus, Ycf48 seems to aid in the assembly of Psb27-bound apo-PSII, instead of the mature PSII. On the other hand, the crashes of mature CP47 and CP43 with Ycf48 seems to indicate the necessity of Ycf48 in preventing the unwanted folding and assembly of CP47 and CP43 before they are matured and incorporated into PSII correctly. This needs to be discussed explicitly.

We agree with the reviewer that we could have been clearer about how Ycf48 affects assembly of the Mn cluster.

Author action: On line 240, we have amended the sentence so that it now reads ‘Our structural data therefore suggest that one role of Ycf48 is to prevent the premature binding and oxidation of Mn ions during PSII assembly so that the light-driven assembly of the cluster takes place at the appropriate stage of PSII biogenesis after attachment of CP47 and CP43.’

On line 269, we have now discussed the role of Psb27.

The text reads ‘Comparison of RCII with the structure of a PSII assembly complex containing both CP47 and CP43, the latter attached to Psb27, reveals substantial clashes between the large luminal loop in CP43 and Ycf48 (Supplementary Fig.13b). Together, the binding of CP47 and the CP43/Psb27 complexes help expel Ycf48 from its binding site on D1 to form the Psb27-bound apo-PSII assembly complex (Supplementary Fig. 1). Subsequent light-driven assembly of the Mn₄CaO₅ cluster requires detachment of Psb27 and reorientation of the C-terminal tails of D1 and D2 (Fig. 3c, d, e) (Zabret et al., 2021).’

Line 332: What is FOM?

We have now explained that FOM is an abbreviation of ‘Fluorinated Octyl Maltoside’ (a detergent additive).

Reviewer #3 (Remarks to the Author):

Photosystem II (PSII) found in photosynthetic organisms is fundamental in defining earth’s biosphere. It acts as the gateway for energy from the sun and it releases oxygen that allows for all higher life on our planet. The study of PSII is widely applicable due to its relevance in fields ranging from synthetic solar fuel catalysis to metalloenzyme assembly. A major gap in our understanding of PSII is its biogenesis, which is vitally important because PSII turnover in vivo is rapid which confers resilience to oxidative damage to the complex. Only recently have some assembly intermediates found during PSII biogenesis been thoroughly characterized, primarily due to new structural data that have provided bases for biochemical interpretation. These assembly intermediates have revealed important features of PSII biogenesis, but primarily those that occur in late-stage assembly. A thorough understanding of early-stage biogenesis intermediates have so far remained elusive. This represents a gap in our understanding of this key enzyme.

Here, Zhao et al. reveal the structural basis of an early biogenesis intermediate of PSII. In the PSII holocomplex, water oxidation is catalyzed by a tetramanganese cluster in the active site. In the assembly intermediate described here, the active site is blocked by an assembly factor called Ycf48 which most likely prevents the premature binding of ions that eventually comprise the metal cofactor. Based on the conservation observed in the regions interacting with the active site,

it is likely that this blocking strategy is relevant to all oxygenic phototrophs. This novel insight enhances our understanding of PSII biogenesis, and it is communicated extremely well both in writing and in figures. I strongly recommend the manuscript for publication in Nature Communications pending some minor edits that are listed here:

PSI structure: I cannot tell how the authors have modeled P700 because I do not have the coordinates, but please use two different residues, CLA and CL0. The CL0 is the Chl a prime that has reversed stereochemistry about its 13-2 carbon atom. 5OY0 which the authors have used as a starting model has this incorrectly modeled by two standard Chl a molecules. An example of its proper use is in PDB 6NWA.

Author action: Thank you for alerting us to this most important point. We have now changed the relevant chl to CL0 and redeposited the structures.

Supp Fig. 5: Please add labels to the figure. It would be helpful to designate which parts correspond to RCII and PSI, and the view perspective, such as “stromal side view” or “membrane plane view”.

Author action: As requested by the reviewer, we have revised Supplementary Fig. 5 to make clear which parts correspond to PSI and which to RCII and have added “view from cytoplasmic side” for the RCII complex.

Line 155: I suggest commenting a bit further on the Cl⁻ ions. Are those regions instead filled with something else? For Cl-2, I imagine this space is instead occupied by Ycf48. Are there any features occupying the space of Cl-1?

Although D1-Lys317 is in a similar position to that of the fully functional PSII, the Cl-1 position clashes with the side chain of Ycf48-Arg196 and no obvious anion-like density exists around the expected position of Cl-1. The Cl-2 site is occupied by the backbone of Ycf48-Thr223 in blade 5 of Ycf48.

Although no Cl⁻ ions were identified in the map, anions are especially challenging to resolve in cryo-EM maps (as opposed to X-ray crystallography-derived electron density maps) unless the resolution is very high. This is due to the negative electrostatic potential scattering factors of anions.

An alteration to the text such as “Importantly, there was no indication in the structure for the presence of bound Mn or Ca ions. Additionally, the two chloride ions, called Cl-1 and Cl-2, were not identified in the cryo-EM map, although it should be noted that anions are challenging to resolve at medium resolution in cryo-EM (REF)” would be sufficient. An appropriate citation would be Wang 2017 (<https://doi.org/10.1002/pro.3198>).

Depending on how the coauthors edit this, they may also want to revise the sentence starting on line 262.

Author action: In the revised manuscript, we have added the following text and the citation recommended by the reviewer.

Added text at line 150: ‘Additionally, the two chloride ions, called Cl-1 and Cl-2, required for functionality of the oxygen-evolving complex (Imaizumi & Ifuku, 2022) were not identified in the cryo-EM map although it should be noted that anions are challenging to resolve at medium resolution in cryo-EM (Wang, 2017). The Cl-1 site close to D2-Lys217 in oxygen-evolving PSII clashes with the side chain of Ycf48-Arg196 and Cl-2 is occupied by the backbone of Ycf48-Thr223.’

Line 174/Supp Fig. 3/Supp Fig. 10: Please describe how conservation was determined and cite ConSurf, or at least include a link to it. Were the authors required to input sequences? Were they aligned? If so, how? The authors could add this as a small paragraph in the methods section.

Author action: We have now added details and suitable citations in the materials and methods section plus an additional supplementary table (Supplementary Table 3) to describe how the ConSurf analysis was done.

Added text at line 378: ‘...and the ConSurf analysis of Ycf48 was done according to (Yu et al., 2018) using the ConSurf server at <https://consurf.tau.ac.il/> (Glaser et al., 2003; Landau et al., 2005). Briefly, 382 unique homologs of Syn6803 Ycf48 were collected from UNIREF90 (Uniport) by HMMER with an E-value of 0.0001 or less and a sequence identity between 35% and 95%. 150 sequences (listed in Supplementary Table 3), which is the maximum number that can be analyzed, were aligned by MAFFT, and a conservation score for each residue was assigned from 1 with the lowest score to 9 with the highest score. The residues of Ycf48 were coloured according to their conservation score, from light green (score 1) to deep green (score 9) in the main figures and from turquoise (score 1) to maroon (score 9) in the supplementary figures. Amino-acid positions for which the inferred conservation level was assigned with low confidence were coloured yellow.’

Supp. Fig. 10: There is an incomplete reference. Also, the last part of the sentence doesn't make sense to me. What are the authors trying to describe about ConSurf here?

Author action: The sentence has been edited to remove the unwanted text.

Line 208: Typo – the authors meant to close the parentheses instead of including a semicolon.

Author action: Corrected.

Line 210: I suggest that the authors clarify this sentence. On line 90, the authors stated “Both precursor and mature forms of D1 were detected in the RCII/PSI preparation suggesting a heterogeneous population of complexes.” Therefore, I don't think that only mature D1 forms were imaged during data collection, which is how I read it. Perhaps they mean that the ensemble 3D reconstruction (i.e., the cryo-EM map) only captures the fraction of particles that contain the mature form of D1? The authors may suggest that the mature D1 C-terminus is relatively homogeneous in its position, but that the unprocessed D1 C-terminal extension positions are heterogeneous.

As discussed in response to a similar point made by reviewer 1, one explanation for the lack of the D1 extension in the model is its high flexibility preventing its recognition in the structural map. The other explanation we suggested was that the complexes containing iD1 did not bind to the carbon grid and so evaded detection.

Authors action: In light of the reviewer's comments, we have modified the text starting from line 204:

Added text: ‘This previous work suggests that the D1 extension is capable of binding to Ycf48 so its absence in the cryo-EM structure may indicate that only complexes containing the mature form of the D1 subunit were imaged or that the mature D1 C-terminus is relatively homogeneous, whereas the position of the C-terminal extension is more heterogeneous and flexible.’

Line 216: Does the red algal Ycf48 homolog have a lipid anchor? Note the inverse correlation between binding affinity and lipid anchor discussed for PsbQ and the Psb27 assembly factor reported recently <https://doi.org/10.1007/s11120-021-00888-2>. The authors may want to add a comment accordingly.

The chloroplast homologs of Ycf48, including the red algal ones, do not contain the lipid anchor (see Knoppová et al. 2021). We speculate that the reason might be related to the fact that biogenesis of PSII in cyanobacteria and chloroplasts occurs in different membrane environments (biogenesis centers near the plasma membrane in cyanobacteria and stromal membranes in chloroplasts). As the algal homolog of Ycf48 does not contain the lipid anchor, which stabilizes the protein binding to the membrane, it is possible that the chloroplast protein acquired the extra loop

to stabilize Ycf48 membrane binding by other means.

Authors action: We have now added the following text starting on line 212 :

Added text: As eukaryotic Ycf48 lacks the N-terminal lipid anchor (Knoppová et al., 2021), the chloroplast protein may have acquired the extra loop to stabilize Ycf48 membrane binding by other means.

References

- Bondarava, N., Un, S. and Krieger-Liszka, A. (2007) 'Manganese binding to the 23 kDa extrinsic protein of Photosystem II' Biochim. Biophys. Acta **1767**, 583-588.
- Cao, P., Xie, Y., Li, M., Pan, X., Zhang, H., Zhao, X., Su, X., Cheng, T. and Chang, W. (2015) 'Crystal Structure Analysis of Extrinsic PsbP Protein of Photosystem II Reveals a Manganese-Induced Conformational Change' Molecular Plant **8**, 664-666.
- Glaser F., Pupko T., Paz I., Bell R.E., Bechor D., Martz E. and Ben-Tal N. 2003. 'ConSurf: Identification of Functional Regions in Proteins by Surface-Mapping of Phylogenetic Information' Bioinformatics **19**, 163-164.
- Knoppová, J., Sobotka, R., Yu, J., Konik, P., Halada, P., Nixon, P. J. and Komenda, J. (2014) 'Discovery of a chlorophyll binding protein complex involved in the early steps of photosystem II assembly in *Synechocystis*' Plant Cell **26**, 1200-1212.
- Knoppová, J., Sobotka, R., Yu, J., Konik, P., Nixon, P. J. and Komenda, J. (2016) 'CyanoP is involved in the early steps of photosystem two assembly in *Synechocystis* sp. PCC 6803' Plant Cell Physiol. **57**, 1921-1931. <https://doi.org/10.1093/pcp/pcw115>
- Knoppová, J., Yu, J., Janoušková, J., Halada, P., Nixon, P. J., Whitelegge J. P. and Komenda, J. (2021) 'The Photosystem II assembly factor Ycf48 from the cyanobacterium *Synechocystis* sp. PCC 6803 is lipidated using an atypical lipobox sequence' Int. J. Mol. Sci. **22**, 3733. <https://doi.org/10.3390/ijms22073733>
- Knoppová, J., Sobotka, R., Yu, J., Bečková, M., Pilný, J., Trinugroho, J. P., Csefalvay, L., Bína, D., Nixon, P. J. and Komenda, J. (2022) 'Assembly of D1/D2 complexes of photosystem II: binding of pigments and network of auxiliary proteins' Plant Physiol. **189**, 790-804. <https://doi.org/10.1093/plphys/kiac045>
- Komenda, J., Nickelsen, J., Tichý, M., Prášil, O., Eichacker, L. A. and Nixon, P. J. (2008) 'The cyanobacterial homologue of HCF136/YCF48 is a component of an early Photosystem II assembly complex and is important for both the efficient assembly and repair of Photosystem II in *Synechocystis* sp. PCC 6803' J. Biol. Chem. **283**, 22390-22399.
- Landau M., Mayrose I., Rosenberg Y., Glaser F., Martz E., Pupko T. and Ben-Tal N. (2005) 'ConSurf 2005: the projection of evolutionary conservation scores of residues on protein structures' Nucl. Acids Res. **33**, W299-W302.
- Michoux, F., Takasaka, K., Boehm, M., Nixon, P. J. and Murray, J. W. (2010) 'Structure of CyanoP at 2.8 Å: implications for the evolution and function of the PsbP subunit of Photosystem II' Biochemistry **49**, 7411-7413.
- Rahimzadeh-Karvansara *et al.* (2022) 'Psb34 protein modulates binding of high-light-inducible proteins to CP47-containing photosystem II assembly intermediates in the cyanobacterium *Synechocystis* sp. PCC 6803' Photosynth Res **152**, 333–346. <https://doi.org/10.1007/s1120-022-00908-9>
- Yu, J., Knoppová, J., Michoux, F., Bialek, W., Cota, E., Shukla, M. K., Strašková, A., Aznar, G. P., Sobotka, R., Komenda, J., Murray, J. W. and Nixon, P. J. (2018) 'Ycf48 involved in the biogenesis of the oxygen-evolving photosystem II complex is a 7-bladed beta-propeller protein' Proc. Natl. Acad. Sci. USA **115**, E7824-E7833. doi:10.1073/pnas.180060911

REVIEWERS' COMMENTS

Reviewer #1 (Remarks to the Author):

The authors appropriately addressed the points I had raised.

Reviewer #2 (Remarks to the Author):

The authors have addressed my concerns adequately, and I can now recommend the acceptance of this paper pending the modifications of the following points.

Lines 67-68: "...in an RCI assembly complex that was isolated attached to a monomeric PSI complex". Needs re-phrasing.

Lines 95-100: The descriptions in this part does not match with what is shown in Supplementary Fig. 4. In the Supplementary Fig. 4, the preliminary processing yielded 6 classes, instead of 2 classes stated in the text. Among the 6 classes, 2 (106532 particles) are utilized for the subsequent templates, which yielded 5 classes instead of 2 stated in the text.... These should be described explicitly in the text and/or in the Methods section.

Reviewer #3 (Remarks to the Author):

The authors have adequately addressed my comments and I recommend the article for publication.

REVIEWER COMMENTS

Reviewer #1 (Remarks to the Author):

The authors appropriately addressed the points I had raised.

Reviewer #2 (Remarks to the Author):

The authors have addressed my concerns adequately, and I can now recommend the acceptance of this paper pending the modifications of the following points.

Lines 67-68: "...in an RCII assembly complex that was isolated attached to a monomeric PSI complex". Needs re-phrasing.

Reply: Unfortunately, it is not clear to us which part of the sentence the reviewer would like re-phrasing. We have modified the sentence on lines 67-68 (lines 70-72 in revised manuscript) to 'Here we have used cryo-EM to determine the binding site of Ycf48 in a RCII assembly complex which was isolated attached to a monomeric PSI complex', which we think is unambiguous.

Lines 95-100: The descriptions in this part does not match with what is shown in Supplementary Fig. 4. In the Supplementary Fig. 4, the preliminary processing yielded 6 classes, instead of 2 classes stated in the text. Among the 6 classes, 2 '106532 particles) are utilized for the subsequent templates, which yielded 5 classes instead of 2 stated in the text.... These should be described explicitly in the text and/or in the Methods section.

Reply: We agree with the reviewer and have revised the text to make the analysis clearer:

Revised text for Lines 95-100 (lines 97-109 in revised manuscript): 'We collected 2853 cryo-EM micrographs of the RCII/PSI complex and picked 1.25 million particles using LoG picking¹⁹ for subsequent data processing. The preliminary processing revealed the existence of two good 3D classes out of a total of six, differing by the relative orientation of RCII to PSI (Supplementary Figs. 4 and 5). The particles corresponding to the two classes (106532 particles) were used as references to repeat picking using Topaz²⁰ and this new set of particles was subject to multiple rounds of 3D classification, as well as polishing and refinements. The best particles were finally separated into 3 classes by 3D classification without alignment, resulting in the same two good classes identified initially, but with more particles per class and higher resolution. The final cryo-EM maps had a resolution of 3.2 Å and 3.1 Å for the two final classes. Additionally, all the particles were pooled to generate a consensus PSI map at 2.9 Å. The data processing pipeline is explained in detail in the Methods section and outlined in Supplementary Fig. 4 and Supplementary Table 2 and examples of the modelling shown in Supplementary Figs. 6 and 7.'

Reviewer #3 (Remarks to the Author):

The authors have adequately addressed my comments and I recommend the article for publication.